# Managing African Armyworm Outbreaks in Sub-Saharan Africa: Current Strategies and Future Directions

**DOI:** 10.3390/insects16060645

**Published:** 2025-06-19

**Authors:** Grace Kinyanjui, Kahsay Tadesse Mawcha, Lawrence Nkosikhona Malinga, Kaitlyn Soobramoney, Phophi Ṋethononda, Yoseph Assefa, Chibuzor Onyinye Okonkwo, Dennis Ndolo

**Affiliations:** 1Biopesticides Group, International Centre for Genetic Engineering and Biotechnology, Anzio Road, Observatory, Cape Town 7925, South Africachibuzar.okonkwo@icgeb.org (C.O.O.); dennis.ndolo@icgeb.org (D.N.); 2Department of Biological Sciences, University of Embu, Embu P.O. Box 6-60100, Kenya; 3Department of Plant Sciences, Aksum University, Aksum P.O. Box 314, Ethiopia; 4South African Sugarcane Research Institute, Private Bag X02, Mount Edgecombe 4300, South Africa; lawrence.malinga@sugar.org.za (L.N.M.); kaitlyn.soobramoney@sugar.org.za (K.S.); 5School of Life Sciences, University of KwaZulu-Natal, Private Bag X01, Scottsville 3209, South Africa; 6Department of Zoology and Entomology, Forestry and Agricultural Biotechnology Institute (FABI), University of Pretoria, Private Bag X20, Lynwood Road, Hatfield, Pretoria 0083, South Africa; phophi.nethononda@fabi.up.ac.za; 7Department of Crop Production, Faculty of Agriculture, University of Eswatini, Private Bag 4, Luyengo M205, Eswatini; yosepharm@yahoo.com; 8Department of Biochemistry, Faculty of Basic Medical Sciences, University of Calabar, Calabar 540271, Nigeria

**Keywords:** pest control, migratory pest, innovative tools, integrated pest management

## Abstract

**Simple Summary:**

The African armyworm is a destructive, migratory pest that threatens cereal crops and pasture grasses in sub-Saharan Africa, with recent outbreaks resulting in significant agricultural and economic losses. This review addresses the urgent need for improved management strategies by examining the drivers of frequent outbreaks, including climate change, the pest’s biological traits, and gaps in current control practices. This paper aims to evaluate existing surveillance and control methods—such as chemical, biological, and cultural approaches—and to explore innovative technologies, including predictive modeling, artificial intelligence, *Bt* maize technology, molecular biology, and nanotechnology. This study finds that, while traditional methods offer some control, their limitations necessitate integrated, technology-enabled pest management solutions. It concludes that adopting holistic, regionally coordinated strategies that incorporate farmer education, advanced tools, and updated policy frameworks will significantly improve outbreak response and mitigation. These findings are valuable to society as they provide a roadmap for sustainable pest control, helping to safeguard food security, support farmer livelihoods, and enhance agricultural resilience in the face of escalating climate and pest pressures.

**Abstract:**

The African armyworm *Spodoptera exempta* (Lepidoptera: Noctuidae) is a significant pest that affects cereal crops and pasture grasses in sub-Saharan Africa. This migratory pest causes extensive defoliation, which can result in significant yield losses, particularly in maize. This review focuses on the recent outbreaks of the African armyworm and identifies key factors contributing to its success across sub-Saharan Africa. Understanding these factors is essential for developing effective and sustainable pest management strategies. This review emphasizes the importance of innovative technologies and data-driven approaches in managing pest outbreaks and underscores the need to implement technology-enabled integrated pest management (IPM) strategies to control the African armyworm effectively.

## 1. Introduction

The African armyworm (black armyworm) *Spodoptera exempta* Walker (Lepidoptera: Noctuidae) is a significant migratory pest of Poaceae and Cyperaceae species in sub-Saharan Africa (SSA) [1,2]. Although endemic to the African region, the pest has also been reported in Arabia, Australasia, and Southeast Asia [2,3,4,5]. Figure 1 shows the global geographical distribution of the African armyworm. In South Africa, the Vhavenda people refer to it as *phundulu*, often accompanied by the saying, “*Phindulu ndi a dzama vhuvhiṊ ndi sia ndo ita*”, meaning “Even if I die, I leave destruction behind.” In Afrikaans, it is known as *die kommandowurm* (“command worm”). Documenting these local names is crucial, as they facilitate pest identification and enhance communication among farmers, extension officers, and researchers. Understanding local terminology enables timely reporting and intervention during outbreaks, ultimately improving pest management strategies.

African armyworm larvae feed voraciously on foliage, resulting in significant yield losses for cereal crops, including maize, rice, and millet. Damage to pasture grasses is particularly severe on *Cynodon* and *Pennisetum* species [1]. The pest thrives in tropical and subtropical climates, where warm temperatures and adequate rainfall facilitate its lifecycle. Seasonal migrations enable the pest to spread across large geographical regions, often following the Intertropical Convergence Zone. The rapid surges of the pest population in SSA severely threaten the production of cereal crops and pasture grasses [2,6]. During severe outbreaks, the larvae can also attack plant species within the families Brassicaceae, Solanaceae, Fabaceae, Rosaceae, Zingiberaceae, and Malvaceae [1,2]. In recent times, outbreaks in SSA have become more frequent, raising serious concerns over food security, farmers’ livelihoods, and the economic stability of the affected countries [7,8].

Economic losses caused by the African armyworm outbreaks vary between crops and their stages of development. For instance, larval defoliation on maize was reported to cause yield losses ranging from 9% during the early whorl stage to 100% at the pre-tassel stage [9]. Additionally, up to 92% yield losses were recorded on maize planted in trial fields in Kenya [1]. In Zambia, the African armyworm reduced maize production by 11% in the 2012–2013 cropping season, while losses attributed to this pest in 2017 were estimated to increase to 40% [7]. Crop damage also varies significantly depending on the larval density per outbreak, prevailing climatic conditions, regional differences, and the effectiveness of pest control measures [1]. Damage to pasture grasses often leads to livestock poisoning, potentially causing death and paralysis due to the release of cyanide-like chemicals by grasses infested by the pest [1,2,10].

The increasing food security threats associated with frequent pest outbreaks prompt urgent calls for coordinated responses to mitigate potential damage. This review examines the outbreak dynamics of the African armyworm and factors contributing to the pest’s success in SSA. It also provides various perspectives on improving the management of the pest.

## 2. Outbreaks in Sub-Saharan Africa

Outbreaks of the African armyworm have been reported throughout the SSA region, particularly in East, Central, and Southern Africa, since 1919 [1,2]. Notable outbreaks were later observed in the early 1990s in East Africa and West Africa in 2000 [11]. The primary outbreaks usually arise from low-density populations and are carried by convergent wind flow during the rainy season to suitable habitats, where they are concentrated [12]. Under favorable environmental conditions, primary outbreaks initiate sequences of secondary outbreaks, which spread to new areas in the direction of prevailing winds [1]. The magnitude of outbreaks in the landing areas depends on the source population, while the prevailing environmental conditions influence the severity of infestation and the extent of agricultural impact [1,2].

Recent reports of severe African armyworm outbreaks were observed in Eastern Africa in 2022 and 2023 [11]. In response to these outbreaks, the Food and Agriculture Organization of the United Nations (FAO) launched a project titled *Emergency Support to Manage Outbreaks and Infestation by African armyworm in Eastern Africa* in 2023 to extend support to six Eastern African countries, namely Eritrea, Ethiopia, Kenya, Somalia, South Sudan, and Uganda, to enhance their national capacities in the fight against the pest [13]. In early 2025, pest outbreaks were observed in several Southern African countries, including Botswana, Eswatini, Malawi, Zimbabwe, and South Africa [8].

### Outbreaks in South Africa

South Africa has been experiencing significant outbreaks of the African armyworm, which has affected multiple provinces and raised concerns about crop and pasture damage. In February 2025, outbreaks were reported in the Free State, Gauteng, KwaZulu-Natal, Limpopo, Mpumalanga, and North West provinces [10,14]. Figure 2 shows the provinces where the outbreaks have been reported in South Africa. In the Free State, over 70 cases were reported in districts such as Xhariep (in the towns of Petrusburg, Luckhoff, Jacobsdal, Oppermansgronde, Koffiefontein, and Fauresmith) and Lejweleputswa (in the towns of Boshof, Dealesville, Hertzogville, Hoopstad, and Bothaville) [15,16]. The Mangaung Metro’s Soutpan area was also affected. Other affected areas include Groblersdal (Limpopo), Carolina and Badplaas in Mpumalanga, Pretoria North, and Bronkhorstspruit in Gauteng. In March 2025, infestations were confirmed in districts such as ZF Mgcawu, Pixley ka Seme, and Frances Baard of the Northern Cape province [17]. The Department of Agriculture, Land Reform, and Rural Development (DALRRD) in South Africa, along with provincial authorities, have been actively monitoring the outbreaks and advising farmers to remain vigilant and implement both preventive and curative measures to protect their crops and pastures [10,17].

## 3. Drivers Facilitating African Armyworm Outbreaks

The frequent outbreaks of the African armyworm are driven by a combination of environmental, biological, and human-related factors, which favor rapid population growth, migration, and infestation of new areas [1,7,11]. Multiple factors driving the outbreaks of the African armyworm are shown in Figure 3. Therefore, a clear understanding of these drivers is crucial for informing the employment of effective pest management strategies. A key factor driving the success of the pest’s sporadic outbreaks in SSA is favorable climatic conditions characterized by warm temperatures, sufficient precipitation, and frequent winds [6]. These conditions favor the growth of large populations of the African armyworm, primarily in several source areas in East Africa, which subsequently spread to the north and south of SSA [1,12,18]. Previous studies also indicate that outbreaks are seasonal and tend to occur during the rainy season following a period of drought [19,20].

The changing climate in recent times, associated with increasing global temperatures and changing precipitation patterns, further impacts the pest’s population dynamics [21,22,23], leading to the increased intensity and frequency of the African armyworm outbreaks. This is due to the associated effects of climate change on insect pests, including an increased number of generations, expansion of geographical distribution, and changes in migratory patterns [23,24,25]. Additionally, despite being endemic in SSA with a range of native predators, parasitoids, and pathogens, biological pest control lags greatly due to the inadequate buildup of populations necessary for optimal natural control of the African armyworm outbreaks [1]. The migratory nature of the African armyworm also impedes parasitoid development and prevents their proliferation to levels required for effective pest control [2].

The African armyworm has an inherently high reproductive capacity and rapid development rate, characteristics of r-reproductive strategists [1,26], which, under favorable environmental conditions, contribute to the successful initiation and resurgence of outbreaks. Additionally, the density-dependent polyphenism exhibited by the African armyworm larvae serves as a major adaptive strategy that drives the ecological success of the migrating populations [1,27,28]. Further, the shift between the cryptic solitary phase and the visible gregarious larvae is rapid and unpredictable, coupled with aggressive feeding behavior and accelerated development of the latter and sudden proliferation of large swarms capable of destroying crops and pastures over vast areas [11,27]. Moreover, adult moths emerging from gregarious larvae exhibit powerful traits, including high fecundity, faster growth rates, and great flight potential [1,27], which enable the pest to colonize and rapidly populate new environments with suitable host plants. The accelerated life cycle and increased reproductive output lead to rapid population growth and an increased likelihood of outbreaks [9,11]. The pest’s high migratory power is also associated with wind convergence, which concentrates and drives moths far distances and contributes to dense swarms that cause severe infestations on foliage [1,12].

Another driving factor is the ability of larvae from low-density populations to thrive on grasslands during the dry season and, upon the onset of rains, develop rapidly into primary outbreaks capable of causing extensive feeding damage in a short period [12,29]. Additionally, the extensive coverage of tropical and subtropical grasslands, which supports large larval populations, and the abundance of cultivated host plants serve as key environmental drivers shaping the geographic spread of the African armyworm outbreaks across SSA [7,30]. Among the human-related factors that facilitate outbreaks are ineffective interventions, such as inadequate monitoring and delayed pest alerts [1,31], which can lead to late reporting, delayed response measures, insufficient pest control, and resurgence of outbreaks.

## 4. Management of African Armyworm

The management of the African armyworm involves various strategies to control the pest population and mitigate its impact on agriculture. Currently, the effective management of pest outbreaks hinges heavily on proactive pest control approaches, including accurate monitoring and robust early warning and reporting systems, coupled with timely interventions [6,32]. Considerable efforts have been made in monitoring and surveillance using pheromone-based traps, sampling larvae, and radar technology [1,33,34]. Upon the detection of swarms, the most common response measure is the application of chemical insecticides, for which ultra-low volume (ULV) spraying is highly recommended for controlling the African armyworm and other migratory insect pests [1,35,36]. This involves spraying tiny droplets of concentrated insecticides to achieve efficient coverage of large areas with minimal volumes of the formulated product. The common insecticides used to curb the pest’s outbreaks in South Africa include synthetic pyrethroids and carbamates [37]. Other recommended pest control measures, although not often applied, include biological control and integrated pest management (IPM) [33,38]. Overall, a proactive and informed approach is essential for minimizing the economic damage caused by this pest to crops across affected countries in SSA. Further, numerous efforts are required to integrate the existing practices for African armyworm control, including pheromone and light trapping, community-based forecasting, monitoring, and early warning systems, and cultural, biological, and chemical control methods (Figure 4) [1,33].

### 4.1. Surveillance and Predictive Tools for Early Detection of African Armyworm

The African armyworm is known for its sudden, short-lived outbreaks that can cause devastating crop losses. Therefore, timely and accurate early warning systems are essential for alerting farmers and agricultural stakeholders to the potential of impending outbreaks, enabling proactive pest management and mitigation strategies [31,39]. Given the destructive potential of these outbreaks, understanding the drivers behind population fluctuations is crucial. Using system dynamics modeling in VENSIM—including causal loop diagrams and reinforcing/balancing feedback loops—researchers have simulated the African armyworm population dynamics using data from 1980 to 2023. These simulations underscore the pivotal role of climatic factors, particularly precipitation and humidity, in influencing the pest’s transition between solitary and gregarious phases. The findings demonstrate that variations in these environmental elements, such as precipitation and humidity, significantly affect outbreak patterns across Africa, reinforcing the importance of integrating climate data into early warning systems [11]. In addition, species distribution models (SDMs) can be employed to identify not only regions where the pest has already been reported but also areas with environmental conditions that could support its invasion and potential outbreaks if it were to be introduced [7].

### 4.2. Biological and Cultural Control of African Armyworm

Among the biological control products developed to control the African armyworm populations, there is the baculovirus *S. exempta* nucleopolyhedrovirus (SpexNPV) [40]. It is most effective when applied during the early larval stages (first to third instars) of the African armyworm [41]. Early ground and aerial application of SpexNPV can adequately suppress larval populations [33,41]. Since its efficacy depends on targeting specific developmental stages, timely monitoring is essential before application. This is particularly important for smallholder farmers, such as those with less than 2 ha of land and limited resources who often lack adequate knowledge of the pest’s biology and struggle to identify its life stages correctly. Given that environmental factors influence the efficacy of entomopathogenic viruses, a promising solution has been developed through the use of a novel formulation that encapsulates baculovirus in an ENTOSTAT-TiO_2_ wax coating, effectively shielding the viral DNA from ultraviolet (UV) damage on crop surfaces [42,43]. This innovative approach holds great potential to enhance the stability and longevity of viral biopesticides in the field, improving their performance while potentially lowering application costs.

Previous studies have shown that African armyworm larvae are susceptible to several *Bacillus thuringiensis* isolates [1,44,45,46]. However, field applications of these isolates have been limited. An early deployment of egg and larval parasitoids has proven effective as a preventive control measure for African armyworm outbreaks [38]. These findings highlight the importance of implementing biological control strategies that utilize both microbial agents and arthropods to manage this pest.

Cultural control methods for *Spodoptera* species include scouting, crop rotation, early planting, and intercropping [47]. Despite being time-intensive, physical control methods such as hand-picking and digging trenches help reduce pest populations when implemented promptly. Farmers also dig trenches up to 30 cm in depth, which trap the larvae, to which pesticides are thereafter applied [37,48].

### 4.3. Plant-Derived Compounds for African Armyworm Control

Pesticidal plant extracts provide a sustainable alternative to synthetic pesticides by reducing the risk of insect resistance development and minimizing the health risks to both humans and the environment [49]. The application of aqueous neem seed extract and azadirachtin causes mortality in larval instars of the African armyworm, along with molting abnormalities and the disruption of pupal development [50]. In a separate study, the treatment of final-instar larvae with azadirachtin resulted in adverse effects on oogenesis, reproductive maturation, and fecundity, as well as suppression of protein levels and fat body development in female moths [51]. Despite widespread familiarity with pesticidal plants among smallholder farmers in Africa, their use remains limited [52]. This is mainly due to factors such as limited information on their efficacy and safety, inconsistent performance of plant materials, high costs and complexity of registration processes, and an underdeveloped commercial biopesticides sector [53].

### 4.4. Integrated Pest Management of the African Armyworm

Integrated pest management (IPM) is an informed decision-making process used to determine which combination of strategies offers the most effective and sustainable management of pests. Although there are currently no established IPM strategies for managing the African armyworm, key management practices for other *Spodoptera* species can provide valuable insights and be adopted for controlling this pest. One such example is the use of IPM-focused digital tools, including mobile apps and software platforms, implemented for the monitoring and early warning systems of *S. frugiperda* [54,55,56].

Other IPM strategies recommended for *Spodoptera* species incorporate crop rotation, intercropping, hand-picking larvae, crushing eggs, the use of pheromone and light traps, refuge areas, biocontrol, chemical control, and training and capacity building farmers [48,56,57,58]. Additionally, recent studies emphasize the importance of climate-smart IPM strategies, particularly for the long-term management of climate change-induced agricultural pests. This includes incorporating practices such as climate-informed forecasting, developing efficient early warning and rapid response systems, enhancing surveillance and innovative solutions with the aid of advanced digital tools, integrating agroecological and biological pest control approaches, developing resilient crop varieties, and facilitating engagement between all relevant stakeholders, including policymakers [59,60]. Moreover, community engagement and collaborative networks are crucial for developing effective IPM strategies and successfully managing this pest [56,61].

### 4.5. Challenges in the Current Management of the African Armyworm

The life cycle of the African armyworm, which encompasses density-dependent polyphenism, plays a key role in hindering the effective prediction and management of outbreaks [11]. The unpredictable outbreaks result in significant food losses and negatively impact food security in the affected countries [8]. Additionally, ecological and human factors also play a role in population dynamics, which limits the ability to predict and manage outbreaks efficiently [11]. The changing climate, associated with rising global temperatures and extreme weather events, including hotter and longer heatwaves, drought periods, and heavy, unpredictable precipitation, triggers unexpected pest outbreaks, which are difficult to forecast and control [62]. Typically, migratory insects that are adapted to high temperatures tend to undergo longer migration periods, exhibit shorter developmental periods, and experience increased populations due to climate warming. Additionally, global warming induces changes in migration patterns [63,64,65,66,67]. Furthermore, changes to global rainfall patterns, atmospheric nitrogen deposition, and carbon dioxide concentrations could affect insect development, survival, and reproduction, influencing their distribution [66,68].

The long-term use of chemical insecticides often results in decreased susceptibility to the insecticide due to field-evolved resistance [69,70]. Although no significant insecticide resistance has been reported for the African armyworm, the repeated use of chemicals raises serious concerns regarding human health and environmental implications [71]. The baculovirus SpexNPV is highly recommended for controlling the African armyworm [40]. However, previous studies have reported that, when reared at high densities, the pest exhibits greater resistance to NPV infection [72,73]. This is due to the production of high levels of phenoloxidase enzyme in the hemolymph of gregarious larvae, which is essential for eliciting an immune response and, as a result, increasing pathogen resistance [72]. High production costs also limit the application of SpexNPV as a viable alternative to chemical insecticides. The efficacy of entomopathogenic viruses is known to be influenced by abiotic factors, including temperature, relative humidity, solar radiation, and UV exposure [42]. Farmers also attempt to create homemade remedies by mixing larvae killed by SpexNPV with water, but this method has proven ineffective [74]. Furthermore, smallholder farmers may use the same knapsack sprayer for both chemical pesticides and biopesticides, which could lead to cross-contamination, potentially reducing the efficacy of biopesticides due to chemical residues that can inactivate the microbial agents.

### 4.6. Socio-Economic Consequences

Frequent outbreaks of the African armyworm can have severe socio-economic impacts, affecting not only the agricultural sector but also local communities and economies. This impact is gauged in terms of both production and economic losses, as reported for *S. frugiperda* in Ethiopia [75]. The infestation of maize crops by the African armyworm leads to reduced yields and production losses ranging from 9% to 100% [1,7,9]. Significant yield losses, in turn, can result in food shortages and higher prices, negatively impacting the overall economy [75]. Additionally, the cost incurred by implementing necessary control measures can place significant financial strain on farmers and significantly impact their livelihoods [76,77].

### 4.7. Governance and Policy Frameworks

It is imperative that updated policies and procedures are in place to prevent and control African armyworm outbreaks. In South Africa, pest management is governed by the Agricultural Pests Act (Act No. 36 of 1983). Clear policies and procedures ensure that outbreaks are expeditiously reported and addressed. Additionally, it is important that all relevant information is readily accessible to the relevant parties to facilitate accurate species identification and the timely implementation of control measures. Furthermore, there is a need for updated policy frameworks regarding the African armyworm in South Africa, especially considering the recent advancements in pest management strategies.

As a transboundary insect pest in SSA, the successful management of the African armyworm also requires strengthened regional cooperation to help affected countries mitigate the impact of outbreaks. Moreover, intensified efforts are necessary to harmonize relevant regulations and policies across SSA to improve pest surveillance, monitoring, and management at the regional level [78].

## 5. Advancing Technologies and African Armyworm Management

Recent reports indicate increasing frequency and intensity of African armyworm outbreaks within sub-Saharan African countries [8,11]. This is despite local, national, and regional pest control interventions aimed at combating outbreaks. The alarming trend of increased outbreaks calls for more adaptive strategies and advanced innovations to address the changing migratory patterns effectively.

### 5.1. Innovative Digital Technologies

Predictive modeling emerges as a significant application in managing pest outbreaks by providing information on environmental suitability and potential migratory patterns of the pest. For instance, SDMs characterize and predict the distribution of a species using species distribution data and environmental variables [79,80]. These models are crucial for pest monitoring and can be used to identify areas vulnerable to outbreaks [81,82]. Statistical, machine learning, and system dynamics models have been employed to predict the distribution and population dynamics of the African armyworm [7,11]. While these models have been instrumental in predicting the spatial and temporal dynamics of the pest, the system dynamics model serves as a more powerful tool because it integrates several factors influencing the pest’s population dynamics, including biological and ecological factors, as well as the impacts of intervention strategies [11].

Further, the integration of geospatial technologies, such as satellite remote sensing, global positioning systems (GPS), and geographic information systems (GIS), into pest management programs for the African armyworm presents excellent opportunities to advance pest surveillance and early warning systems [83]. These digital technologies offer valuable spatial information, making the large-scale monitoring of outbreak occurrences more accurate and reliable [83,84].

Additionally, revolutionizing the management of the African armyworm will require the application of artificial intelligence (AI) domains, including machine learning and deep learning, and their integration with innovative technologies such as the Internet of Things (IoT), robotics, data analytics, and other digital information systems [85,86,87,88,89]. Considering the similarity in polyphenic development between the African armyworm and the desert locust, *Schistocerca gregaria*, management strategies for these migratory pests are comparable. Several studies have reported that efficient early warning and monitoring systems based on AI algorithms, remote sensing, and GIS are key to the effective management of *S. gregaria* outbreaks [35,36,90]. Moreover, Gómez et al. [91] demonstrated the use of SDMs based on machine learning algorithms to identify suitable habitats for *S. gregaria*, providing information necessary to inform decision-making, improve existing early warning systems, and prevent uncontrollable outbreaks.

Artificial intelligence offers the potential to enhance surveillance and facilitate the real-time and precision monitoring of pest populations. Artificial intelligence models can assist in the prevention of outbreaks by analyzing historical data to predict when and where future infestations are likely to occur [11]. This can help farmers implement control measures when necessary. Effective strategies for managing outbreaks can also be created and improved by utilizing AI models to simulate various outbreak scenarios [7]. With the aid of AI, the ability to accurately identify pest species is enhanced, thereby enabling early detection, which is crucial for effective control [92]. Furthermore, integrating AI-driven technologies into geospatial technologies and predictive modeling facilitates timely and informed decision-making, as well as immediate interventions in managing the pest [89]. Implementing early warning systems also ensures efficient resource management and minimizes economic loss [31].

### 5.2. Genetically Modified Bacillus thuringiensis (Bt) Maize

Genetically modified *Bt* maize events expressing crystal (Cry) proteins have successfully been used to control various lepidopteran pests. In South Africa, two *Bt* maize events are cultivated for the control of maize stem borers, *viz*., MON810 (expressing Cry1Ab) and MON89034 (expressing Cry1A.105 + Cry2Ab2) [93]. In a previous study, the first and third instars of the African armyworm, when feeding on MON810 and MON89034 maize, exhibited reduced survival and mean larval weight, with the MON89034 maize event showing higher efficacy [94]. However, 6% of third instar larvae that fed on MON810 maize survived and successfully pupated. These studies demonstrate the applicability of *Bt* maize technology for controlling the pest. A significant challenge to using *Bt* maize is the development of field-evolved resistance. This challenge underscores the importance of proactive resistance management strategies, such as incorporating refuge areas [95]. For the management of *S. frugiperda* in Sub-Saharan Africa, a minimum refuge of 50% and 20% non-*Bt* maize is recommended for maize containing one or two effective toxins against the species, respectively. It is also recommended that the next generation of *Bt* maize contains four distinct toxins and a minimum refuge of 10% non-*Bt* maize [96]. In addition to Cry proteins, transgenic *Bt* maize events that produce vegetative insecticidal proteins (Vips), particularly Vip3Aa20, are promising candidates for controlling the African armyworm [97,98]. Previous studies have demonstrated the insecticidal activity of Vip3A proteins against various *Spodoptera* species, including *S. frugiperda*, *S. exigua*, and *S. litura* [98,99,100], highlighting the necessity to evaluate these proteins’ toxicity on African armyworm populations present in different geographical locations across SSA. Furthermore, it is important to explore the potential synergy between Cry and Vip3A proteins to enhance their insecticidal effectiveness against this significant migratory pest [97,98,101]. Importantly, the implementation of *Bt* maize technology should be considered as part of IPM strategies to improve the control of the African armyworm [96,97].

### 5.3. Sterile Insect Technique

The sterile insect technique (SIT) aims to reduce pest populations by releasing mass-reared sterile insects into the wild. The sterile males then compete to mate with wild females, resulting in no offspring and thus reducing reproduction and population size [102]. Classical SIT programs use gamma irradiation to sterilize the males. Although the technology offers an environmentally sustainable, species-specific, and resistance-free pest management strategy [102], a significant research gap exists regarding its use in controlling the African armyworm. However, previous studies have demonstrated success with other *Spodoptera* species [103,104,105]. For instance, a recent study successfully determined an optimal radiation dose and overflooding ratio in *S. frugiperda* [106]. Similarly, high sterility and mating competitiveness were achieved in *S. littoralis*, thus indicating the effectiveness of SIT as a management strategy for *Spodoptera* species [107].

### 5.4. Molecular Technologies in the Management of African Armyworm

Molecular technologies are increasingly used to advance pest management through insect taxonomy, the analysis of plant–insect interactions, host-plant resistance breeding, genetic transformation to improve SIT, genome editing and gene silencing methods, and other genetic engineering approaches [108,109,110,111]. For instance, advances in genetic engineering have provided new opportunities to improve the performance of sterile males released into the wild and enhance SIT through transgenic technology, gene editing, and RNA interference (RNAi) [112,113]. These studies could open new avenues for improving the control of African armyworm populations.

Exploiting RNAi by triggering the double-stranded RNA (dsRNA) molecules [114] serves as a promising strategy to control the African armyworm. This biotechnological approach has been explored in several insect pests, including *Spodoptera* species [111,115,116,117,118], although there are no reported studies on the use of RNAi in the African armyworm. Therefore, research should emphasize the application of RNAi-based technologies as a tailored approach to managing the pest. Furthermore, the successful application of an RNA-based strategy will require addressing, among other challenges, the timely and efficient uptake of dsRNA by early instar larvae, as well as the high costs of dsRNA production [119].

The clustered regularly interspaced short palindromic repeats/CRISPR-associated system is a gene editing tool with potential applications in suppressing pest populations [120]. This can be achieved using gene drive to target genes essential to the pest’s survival or fertility, thereby reducing population numbers over time [121]. Genome editing can also reduce or eliminate pesticide resistance, usually through gene knockout, thus making insect pests more susceptible to insecticides [122]. The CRISPR/Cas9 system has been successfully used in *Spodoptera* species. For instance, previous studies have demonstrated that mutagenesis was effectively induced in *S. litura* [120] and *S. frugiperda* [123] by targeting the developmental gene, homeotic *abdominal-A* gene. Another study on *S. litura* demonstrated that males exhibited a reduced response to sex pheromones following mutation of the *SlitPBP3* gene using the CRISPR/Cas9 system [124]. Additionally, gene knockout was used to investigate the function of *Orco*, an olfactory receptor gene, in *S. littoralis*. This study found that *Orco* knockout resulted in the inability to detect sex pheromones and plant odor [125]. The CRISPR/Cas9 technology also offers a tailored approach to controlling the African armyworm and mitigating its detrimental impact on food and feed production. Therefore, research is needed to modify the genome of the pest and disrupt its growth and development, influence its mating behavior, or induce sterility as observed in other *Spodoptera* species. Future recommendations also include integrating the CRISPR/Cas9 system with strategies such as SIT to enhance pest management [126].

### 5.5. Nanotechnology

Nanotechnology presents significant potential for the more effective, sustainable, and environmentally conscious management of African armyworm outbreaks by offering innovative solutions across various aspects of pest control. Chemical and biological pesticides can be formulated through nanoparticle encapsulation for enhanced delivery and controlled release [127,128,129], improving pesticide efficacy and reducing off-target effects. Additionally, nanoformulations enhance solubility and uptake [130] and increase pesticide toxicity via nanoemulsions with superior infiltration capabilities [131]. Previous studies have shown that nanopesticides have great potential in controlling lepidopteran species [132,133,134,135,136,137,138]. For instance, the application of zinc oxide nanoparticles induced body malformation and reduced fecundity in *S. frugiperda* [139]. Another study reported 98% and 90% mortality in *S. litura* larvae treated with silver nitrate and zinc oxide nanoparticles, respectively [140].

Various nanomaterials, including nanocapsules, nanogels, nanoparticles, and nanoemulsions, are utilized to formulate and deliver biopesticides, such as plant extracts or microorganisms, thereby improving their stability, longevity, water solubility, and field efficacy [139,141,142,143]. For example, silver nanoparticle formulations of plant extracts have exhibited notable larvicidal activity against *S. frugiperda* larvae [130,144,145]. Beyond delivery systems, nanosensors can enable the early detection of insect pests or plant stress signals, facilitating timely interventions to prevent extensive outbreaks [130]. Additionally, nanoparticles, like zinc oxide, can exhibit direct insecticidal properties by affecting pest development and life cycles [133,139]. Additionally, nanomaterials can be engineered to disrupt insect physiology or behavior through novel mechanisms [130]. Recent studies are increasingly integrating nanotechnology with RNAi to enhance the delivery of dsRNA using nano-enabled delivery systems, thereby contributing to the improved efficiency and stability of RNAi-based pesticides [118,146,147].

However, despite several advantages, including targeted delivery, controlled release mechanisms, and enhanced efficacy against insect pests, the application of nanotechnology in pest management lags in SSA [148,149]. Further, realizing these benefits necessitates the careful consideration of potential risks, including the environmental fate and toxicity of nanoparticles [150]. Once introduced into the agricultural environment, nanoparticles can undergo a series of complex interactions and transformations that determine their fate and transport within soil and water systems [151,152]. Concerns have also been raised regarding the potential for some nanoparticles to exhibit toxic effects on non-target organisms in the agricultural ecosystem [153]. Hence, further research specifically tailored to the African armyworm and the African agricultural context is crucial for optimizing and ensuring safe application. Importantly, addressing certain limitations, including the high cost of nano-based technologies, production scalability, ecotoxicity concerns, and other regulatory hurdles [149], is crucial to the successful development of nanoformulations for pest control.

## 6. Technology-Enabled Integrated Pest Management Strategies for African Armyworm

Effective IPM for the African armyworm, while facing challenges due to increasing outbreak frequency and intensity [7,8], necessitates a holistic approach that integrates conventional IPM practices with innovative technologies, such as advanced digital tools, AI, data analytics, predictive modeling, molecular tools, biotechnology, and nanotechnology [7,11,32,78,83,102,111,128,129]. A holistic pest control approach also focuses on proactive measures to prevent pest problems, including monitoring, surveillance, and early warning systems, as well as reactive measures to reduce pest-induced crop losses, such as responsible pesticide use, adoption of biological control, and other sustainable farming practices. A typical technology-enabled IPM approach could employ the application of sensor-based trap systems, drones, statistical and machine learning approaches, and IoT for automatic pest detection and monitoring, coupled with *Bt* maize technology and the drone-based spraying of SpexNPV and *B. thuringiensis* as biopesticides, as well as synthetic pyrethroids [38,86,154]. Drone technology plays a crucial role in managing migratory insect pests by facilitating the ULV application of insecticides and enabling surveillance to detect the presence and density of swarms [36,90,155].

Innovative technologies have the potential to transform conventional IPM programs by introducing new tools and data-driven approaches that will facilitate timely interventions and prevent the large-scale destruction of crops and pastures. However, the successful implementation of technology-enabled IPM programs in SSA will require addressing several challenges, such as high costs of innovative technologies, limited accessibility, availability, and affordability, scaling-up constraints, and low uptake by end-users. Strengthening regional cooperation among countries in SSA, particularly through cross-border collaborations, could overcome several challenges by fostering a collective commitment to drive innovations, sharing necessary information, expertise, and resources, and implementing coordinated actions, such as pest surveillance, monitoring, and control. Additionally, continued research is essential to advance African armyworm management, coupled with the development of appropriate regulatory frameworks and a favorable environment that enables innovative technologies. Other critical factors include adequate knowledge and resources to manage the pest, effective planning, timely decision-making, and good communication tools for sharing information on potential outbreaks and disseminating IPM technologies [156]. Importantly, the success of technology-enabled IPM strategies against the African armyworm heavily relies on well-informed farmers. This requires integrating farmers’ training to improve their knowledge of surveillance and pest management and to empower them in community-based forecasting. Additionally, training programs and the sharing of relevant resources are crucial for empowering farmers with the knowledge to interpret early warning systems derived from geospatial technologies [157], to understand the principles behind AI-driven monitoring and intervention strategies [7,11], and to adopt sustainable practices arising from molecular and nanotechnological advancements. Without comprehensive training, the potential benefits of these innovations in achieving more adaptive and effective IPM strategies for the African armyworm will be significantly limited.

## 7. Conclusions

The increasing outbreaks of the African armyworm in sub-Saharan Africa continue to pose a significant threat to agricultural productivity and food security. This necessitates innovative approaches to fight the pest and reduce its impact on cereal crops and pasture grasses. By integrating technological innovations with conventional methods, the improved control of pest outbreaks can be achieved. Further, addressing the barriers to the successful application of advanced pest control technologies is crucial for harnessing the full potential of technology-enabled IPM strategies against this pest to drive agricultural productivity and food security in SSA. Enhancing regional collaboration and farmer education is also crucial in reducing the impact of this pest across SSA.

## Figures and Tables

**Figure 1 insects-16-00645-f001:**
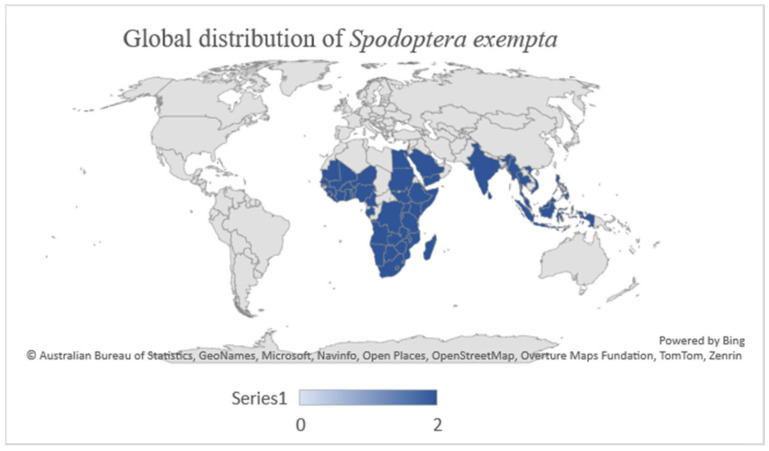
Global distribution of the African armyworm. Countries highlighted in blue show the presence of the pest.

**Figure 2 insects-16-00645-f002:**
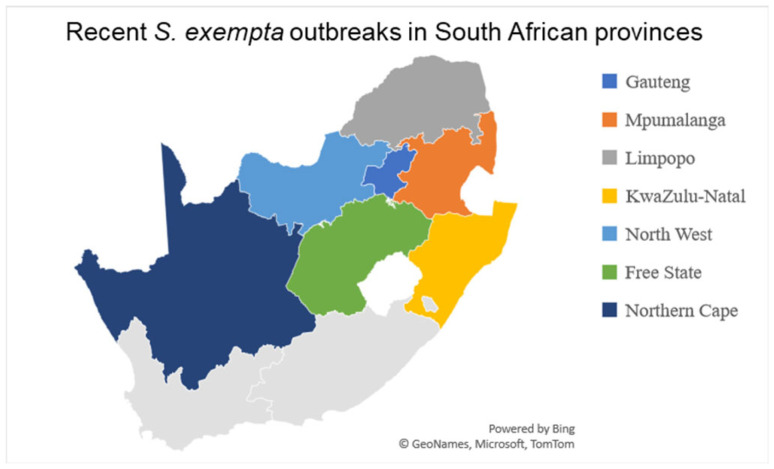
Recent outbreaks of African armyworm in South African provinces.

**Figure 3 insects-16-00645-f003:**
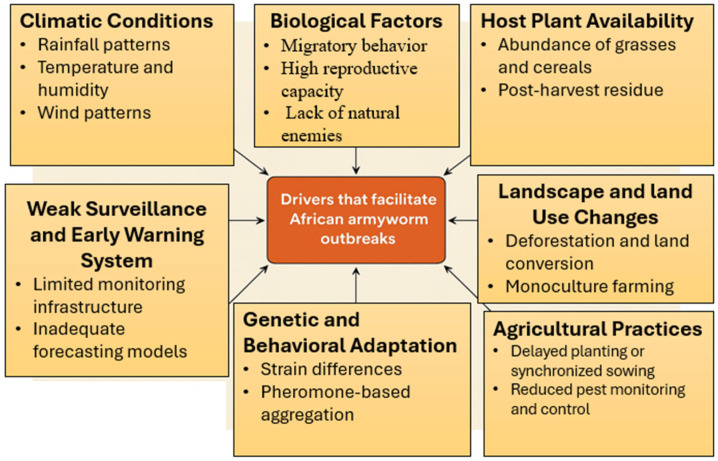
Drivers of African armyworm outbreaks in sub-Saharan Africa.

**Figure 4 insects-16-00645-f004:**
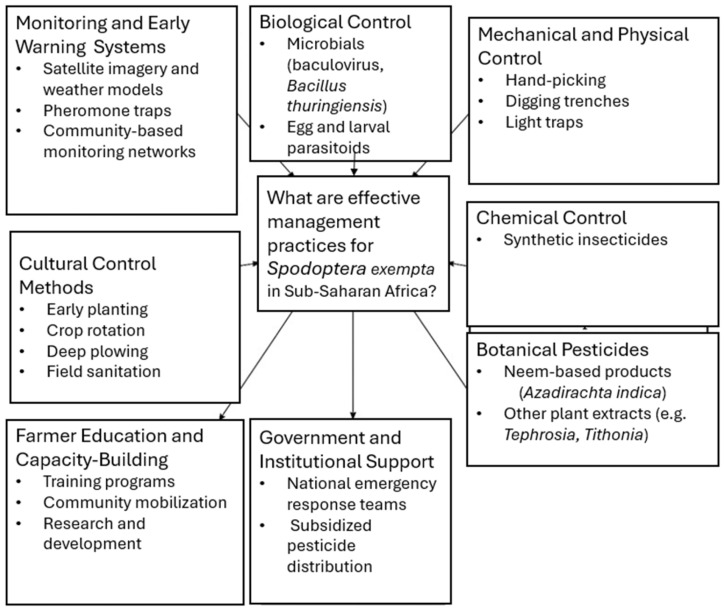
Management practices for the African armyworm in sub-Saharan Africa.

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
