# Peer review of "Managing African Armyworm Outbreaks in Sub-Saharan Africa: Current Strategies and Future Directions"

_insects, 2025, doi:10.3390/insects16060645_

Round 1
Reviewer 1 Report
Comments and Suggestions for Authors
African armyworm is a highly aggressive agricultural invasive pest found across various global regions such as Sub-Saharan Africa, Oceania and Southeast Asia. This review focuses on the recent outbreaks of the African armyworm in Sub-Saharan Africa. It highlights the key factors driving its success across sub-Saharan Africa, which are crucial to achieving effective and sustainable management of this pest. The review underscores the role of innovative technologies and data-driven approaches in managing pest outbreaks while emphasizing the need to leverage technology-enabled integrated pest management strategies in African armyworm control. It also would be valuable for the potential invasive countries to manage the pest.
However, it requires further improvements before it can be accepted for publication.
- Figure 4. Biological Control needs show in the figure. And in the frame of chemical control in Figure 4, botanical pesticides and other plant extracts were listed as chemical control, which should be listed in biological control frame,
- In Advancing technologies and African armyworm management, Bt maize should be included. Bt maize can manage many lepidopteran insect pests, such as fall armyworm.
- Sterile insect technique does not belong to molecular technology. It should be listed as Sterile insect technology, separately.
Author Response
Reviewer 1
Comments and Suggestions for Authors
African armyworm is a highly aggressive agricultural invasive pest found across various global regions such as Sub-Saharan Africa, Oceania and Southeast Asia. This review focuses on the recent outbreaks of the African armyworm in Sub-Saharan Africa. It highlights the key factors driving its success across sub-Saharan Africa, which are crucial to achieving effective and sustainable management of this pest. The review underscores the role of innovative technologies and data-driven approaches in managing pest outbreaks while emphasizing the need to leverage technology-enabled integrated pest management strategies in African armyworm control. It also would be valuable for the potential invasive countries to manage the pest.
However, it requires further improvements before it can be accepted for publication.
Response: Yes, we have carefully considered the reviewer’s comments and revised the manuscript.
Figure 4. Biological Control needs show in the figure. And in the frame of chemical control in Figure 4, botanical pesticides and other plant extracts were listed as chemical control, which should be listed in biological control frame.
Response: Yes, this has been done. Biological control is included in Figure 4. Botanical pesticides are separated from Chemical control.
In Advancing technologies and African armyworm management, Bt maize should be included. Bt maize can manage many lepidopteran insect pests, such as fall armyworm.
Response: Yes, this has been done. Lines 381-398. We have included a section on “Genetically Modified Bacillus thuringiensis (Bt) maize”
Sterile insect technique does not belong to molecular technology. It should be listed separately as Sterile Insect Technology.
Response: Yes, this has been done. Lines 399-410. The section on “sterile insect technique” is separated form molecular technologies.
Reviewer 2 Report
Comments and Suggestions for Authors
See attached file.

Author Response
Reviewer 2
This is an authoritative, timely, up-to-date and well-prepared review. Apart from two general comments that the authors may wish to consider, there are only a small number of minor corrections (noted below).
Response: We have carefully considered the reviewer’s comments and revised the manuscript.
General comments: in reading about the population dynamics and migratory spread of S. exempta, I am immediately reminded of the biology of the desert locust Schistocerca gregaria. Are there major similarities in biology and management of the two species that merit some comment on the comparison? Are there lessons from locust pest management that can, or already are being applied to the armyworm?
Response: Lines 360-368. We have included that these two pests are migratory and exhibit polyphenism. We have included studies on the management of Schistocerca gregaria that are applicable to the African armyworm.
- Lines 503-505, we have stated the role of drone technology in managing migratory insect pests.
In a similar vein, how does the situation with the recently introduced S. frugiperda compare to that for S. exempta? What biological aspects do the two species share, or alternatively, how do they differ? Are there common tools/strategies for pest management that are applicable to both species? You mention some of the microbial pesticides and the potential for RNAi, but how about some current IPM practices?
Response: Lines 256-275. We have included a section on IPM of the African armyworm. The section presents management practices for other Spodoptera species that can be adopted for the African armyworm.
Minor corrections:
Line 16- delete ''foliage'
Response: Yes, this has been done. Line 35. The word “foliage” was deleted between the words “extensive” and “defoliation”
Line 213 – change 'timeously' to 'in a timely fashion'
Response: Yes, this has been done. Lines 240-241. The words “when done in a timeously” were replaced with the words “when implemented promptly”
Line 217 - change 'evolution' to 'development'
Response: Yes, this has been done. Line 245. The word “evolution” was replaced with “development”
Line 272 - italicize S. exempta
Response: Yes, this has been done. Line 321. The word S. exempta was replaced by African armyworm.
Line 380 - 'lepidopteran' (no capital L when used as an adjective)
Response: Yes, this has been done. Lines 457. The word “lepidopteran” is not capitalized.
Figure 1 – I appreciate that you are showing countries on a global scale, but it seems senseless
to include Alaska. Yes, it is part of the United States, but it is far too cold for any Spodoptera species to survive and thrive there.
Response: The map was edited to exclude USA and Alaska for countries showing the presence of the African armyworm.
Figure 3 - something seems missing from the bottom of the "Agricultural Practices" box.
Response: Yes, this has been done. Figure 3 was edited to include the missing information.
Figure 4 - 'Satellite Imagery" - reduce font size for consistency; 'deep plowing' and 'Botanical pesticides' - remove colon; Tephrosia, Tithonia – italicize
Response: Yes, this has been done.
References - italicize ALL Latin binomials and names of genera
Response: Yes, this has been done. Lines 550-893. Latin binomials and genera are italicized.
Round 2
Reviewer 1 Report
Comments and Suggestions for Authors
The revision is fine. As vegetative insecticidal proteins (Vips) are more active to Spodoptera species than Cry proteins, such as MIR602 expressing vip3Aa20 protein is more toxic to fall armyworm than Cry1Ab and Cry1Fa. vip toxins and related events should be introduced in Bt maize part.
Author Response
Reviewer
Comments for Authors
The revision is fine. As vegetative insecticidal proteins (VIPs) are more active against Spodoptera species than Cry proteins, such as MIR602 expressing the vip3Aa20 protein, they are more toxic to fall armyworm than Cry1Ab and Cry1Fa. vip toxins and related events should be introduced in the Bt maize part.
Response: Yes, this has been done. Lines 396-405. We have included the applicability of vegetative insecticidal proteins in controlling the African armyworm.